# The Impact of the COVID-19 Pandemic Caused by SARS-CoV-2 on Hygiene, Health, and Dietary Habits: A Survey-Based Study

**DOI:** 10.3390/epidemiologia6040067

**Published:** 2025-10-22

**Authors:** Aleksandra Wdowiak-Szymanik, Katarzyna Grocholewicz

**Affiliations:** 1Independent Pediatric Dentistry Department, Pomeranian Medical University, 70-111 Szczecin, Poland; 2Department of Interdisciplinary Dentistry, Pomeranian Medical University, 70-111 Szczecin, Poland; katarzyna.grocholewicz@pum.edu.pl

**Keywords:** pandemic COVID-19, oral hygiene, dietary habits, dental check-up

## Abstract

**Background:** The COVID-19 pandemic significantly disrupted various aspects of daily life, including hygiene routines, dietary habits, and access to dental care. This study aimed to evaluate the impact of the pandemic on the oral health-related and dietary behaviors and dietary pattern of patients from the West Pomeranian region of Poland. **Methods:** A cross-sectional survey was conducted among 100 healthy adult participants from the West Pomeranian region, including patients from the Pomeranian Medical University and private dental practices. A self-administered questionnaire consisting of 43 items was used to assess changes in hygiene, dietary behaviors, and the frequency of dental visits during the pandemic. **Results:** The majority of respondents were under 30 years of age, with women representing 56% of the sample. Most participants resided in large urban areas with populations exceeding 300,000. During the pandemic, 41% of participants maintained regular dental visits, while 37% reported experiencing dental problems; all those who sought care received appropriate treatment. Nearly half of the respondents had undergone quarantine due to SARS-CoV-2 exposure, and 38% expressed fear of infection. The results revealed a notable decline in preventive dental care during the pandemic: only 41% of participants reported maintaining regular dental check-ups. Additionally, 34% reported increased consumption of snacks, while 25% indicated more frequent alcohol intake. 22% of respondents experienced involuntary teeth clenching during the day, and 13% reported teeth grinding, These findings reflect a negative shift in health behaviors during the COVID-19 period. **Conclusions:** The pandemic had a substantial adverse effect on oral health behaviors, dietary choices, and the use of dental services. Nevertheless, participants demonstrated awareness of these changes and, following the pandemic, expressed an increased understanding of the importance of regular dental visits. It is necessary to implement preventive measures that increase awareness of the health consequences (such as dental caries and periodontal diseases) in order to reduce the neglect of routine dental check-ups.

## 1. Introduction

The SARS-CoV-2 virus, commonly known as the coronavirus responsible for COVID-19, was first identified in Wuhan, China, at the end of 2019. It rapidly spread across the globe, causing severe disruptions to healthcare systems. Due to its high transmissibility and elevated mortality rates, the World Health Organization declared a global public health emergency (PHEIC) and a pandemic on 30 January 2020 [1,2,3,4].

SARS-CoV-2, a member of the coronavirus family, is primarily transmitted through respiratory droplets and demonstrates a high potential for rapid human-to-human transmission. The clinical manifestation of COVID-19 varies considerably among individuals, influenced by factors such as age, general health status, and immune system function [5,6,7].

In response to the pandemic, healthcare systems around the world underwent profound transformations [8,9,10]. Medical institutions–including hospitals, clinics, dental offices, and other healthcare facilities–were compelled to modify their protocols to adapt to the new conditions [8,9]. Institutions that had previously operated efficiently became overwhelmed, and healthcare delivery standards were significantly altered.

To curb the spread of the virus and mitigate rising mortality rates, governments across the globe implemented stringent public health measures. In many countries, full lockdowns were introduced, halting economic activity and enforcing home isolation [10,11,12]. These restrictions limited access to basic healthcare services, while remote work became the norm [13]. Public institutions such as schools, universities, and shopping centers were closed [14,15,16].

Pandemic-related lifestyle changes also affected dietary habits. A growing number of individuals turned to catering services offering health-conscious meal plans or relied on takeout options, while others began preparing meals at home [17].

The global lockdown contributed to heightened stress and social tension, leading to a surge in mental health issues [18,19,20,21,22,23]. Many individuals reported a decline in mental well-being, attributed to enforced isolation, remote work, and restrictions on mobility [24,25,26,27].

Overall health also deteriorated due to reduced access to essential medical procedures, therapies, and treatments. The abrupt shift in daily routines negatively impacted health-promoting behaviors, which were often neglected or significantly altered [18,19,28].

Oral health was similarly affected. Many patients reported worsening dental conditions due to limited access to dental care during the pandemic [29]. A number of dental practitioners reduced in-office services due to infection risk, while many patients avoided routine visits or postponed planned treatments out of concern for their safety [29,30,31].

Although numerous studies have explored the psychological and systemic health consequences of the COVID-19 pandemic, relatively few have specifically examined its effects on oral health and hygiene behaviors at the community level. This study aims to address this gap by investigating how adult patients in Poland’s West Pomeranian region adapted their health-related routines during the pandemic.

The aim of the study was to gather information on the impact of the pandemic on hygiene, dietary, and health habits in order to develop a strategy for action and widely accessible health education in the event of a future pandemic and lockdown. The study also considered the introduction of clear, widely accessible recommendations regarding nutrition, hygiene, and preventive dental measures, aimed at reducing the long-term consequences of neglect in these areas.

**Hypothesis** **H1:***The COVID-19 pandemic and associated lockdown measures have exerted a negative impact on dietary, health-related, and hygiene habits*.

## 2. Material and Methods

A cross-sectional survey study was conducted between October 2020 and October 2022. The study included a group of 100 generally healthy, adult patients, from Pomeranian Medical University and also private dental practices, residing in the West Pomeranian Voivodeship.

The inclusion criteria were as follows: age between 18 and 70 years, complete or partially preserved natural dentition.

Only healthy patients without systemic diseases were qualified for the study, motor disability was not an exclusion criterion. Questionnaires in which patients indicated the presence of general diseases were rejected and not included in the study.

The study received approval from the Bioethics Committee of the Pomeranian Medical University in Szczecin (Approval No. KB-0012/22/2021, dated 28 June 2021) and was conducted in accordance with the guidelines of the Declaration of Helsinki.

The study was based on a patient-completed questionnaire. All participants received detailed information about the study prior to participation and provided written informed consent. The study information included an explanation of the subject and objectives of the study, potential benefits for participants, and the possibility of withdrawing consent at any time. The estimated completion time was approximately 20 min. The data and survey results were recorded and collected solely for the purposes of this study. To ensure respondent anonymity, no identifying information was collected. Questionnaires that were incorrectly completed by participants were excluded from the analysis.

The questionnaire consisted of 43 questions concerning socio-demographic data, hygiene practices, dietary behaviors, and comparisons of behaviors before and during the pandemic. The questionnaire was designed as an open-ended, paper-based survey written in Polish. Some questions allowed for multiple responses.

The survey consisted of two sections. The first gathered socioeconomic information, while the second focused on health, hygiene, and dietary behaviours.

The first section included questions on age, gender, education, health status, medications taken, place of residence, occupation, and type of employment.

The second section explored health-related habits, distinguishing between behaviours before and during the pandemic. It included questions on self-assessed oral health and hygiene. Further questions focused specifically on dental visits, addressing concerns about SARS-CoV-2 infection during appointments, reasons for missing routine check-ups, the need for urgent visits (e.g., due to pain), whether respondents sought professional help, whether they received care, and the emergence of other symptoms affecting the masticatory system.

Additional questions examined daily oral hygiene practices, including frequency and duration of tooth brushing, type of toothbrush used (multiple choice), type of toothpaste, and use of other oral hygiene products.

Dietary habits were assessed through questions on eating patterns, self-perceived quality of diet, the impact of the pandemic on dietary behaviour, frequency of snacking, meal quality, alcohol consumption, and use of stimulants.

The sample size was adjusted to obtain 100 correctly completed questionnaires, after excluding those that met the exclusion criteria or were filled out incorrectly. Prior to the main study, a pilot study was conducted on a group of 20 individuals; the results of this pilot study allowed for clarification and verification of the clarity of selected questionnaire items. The research procedure was standardized: all respondents received the same set of questions, and the survey was conducted by a single interviewer.

### 2.1. Sample Size Determination

A priori power analysis was conducted to determine the minimum sample size necessary for detecting a medium effect size (Cohen’s w = 0.3) in a chi-square test of independence with 4 degrees of freedom, at a significance level of α = 0.05 and statistical power of 0.80. Employing the pwr package in R, the calculation yielded a required total sample size of approximately 133 participants. Given the exploratory nature of the study and resource constraints, a reduced power of 0.70 was also considered, resulting in a minimum sample size of approximately 109 participants.

### 2.2. Statistical Analysis

The data obtained from the survey were subjected to statistical analysis. A significance level of α = 0.05 was adopted. Accordingly, results with *p* < 0.05 were considered to indicate statistically significant relationships between variables. When the conditions for applying the chi-square test were not met, Fisher’s exact test for count data was conducted with simulated *p*-value using Monte-Carlo method (based on 10,000 replicates). The post-hoc pairwise comparisons were performed using Fisher’s exact test, with *p*-values adjusted via the false discovery rate (FDR) correction to account for multiple comparisons. Cramér’s V and phi coefficients were reported as measures of effect size. Calculations were performed using R statistical environment version 4.3.3, PSPP software version 1.6.2, and MS Office 2019.

## 3. Results

The largest group of respondents were patients under the age of 30, with women comprising 56% of the total sample.

Most participants reported living in cities with populations over 300,000, while the smallest share resided in rural areas or towns with fewer than 20,000 inhabitants. More than half had completed secondary education; slightly fewer held college degrees, and the remainder reported secondary or lower secondary education. The majority of respondents were employed, with students making up approximately one-quarter of the group (Table 1).

The majority rated their overall health as “very good,” followed by those who described it as “good.” About 10% assessed their general health status as “moderate.”

Regarding the types of toothbrushes used, 14% of respondents reported using sonic toothbrushes, whereas nearly twice as many (32%) indicated the use of electric toothbrushes. The majority of participants used manual toothbrushes 68%.

More than half of the respondents replaced their toothbrush or the head of an electric/sonic toothbrush on average 3–4 times per year. Six percent of respondents replaced it once a year (Figure 1).

In the question concerning the use of additional oral hygiene products, respondents could select multiple answers (Figure 2). Dental floss and mouthwash were selected with similar frequency. Additionally, 30% of participants reported using chewing gum. A considerable proportion of respondents indicated that they did not use any supplementary oral hygiene products.

The vast majority, 74% of respondents, used fluoride toothpaste, 7% without fluoride, while 19% reported not knowing whether their toothpaste contained fluoride.

Nearly half of the respondents declared having been quarantined due to SARS-CoV-2 infection, whereas the presence of the virus was confirmed in only 34% of participants. Fear of SARS-CoV-2 infection was expressed by 38% of respondents, while 8% were indifferent, and 54% reported no fear of infection.

Regarding concern about contracting SARS-CoV-2 in a dental office, 20% expressed worry, 7% reported an indifferent attitude, and 73% were not concerned about infection.

During the pandemic, 37% of participants experienced a “dental problem,” and of these, 37% sought assistance from a dentist. All patients who sought help received appropriate dental care. A total of 41% of respondents regularly attended dental appointments during the pandemic, 50% visited due to a specific need, 2% visited very rarely, and 7% did not visit a dentist at all.

In response to a question about returning to regular dental check-ups, 76% confirmed they would return to routine visits, 10% could not specify a date for resuming appointments, 6% declared an intention to return once the pandemic ends, 3% were unsure whether they would resume visits at all, and 5% were waiting for improved financial conditions.

When asked about reasons for discontinuing regular dental visits, 21% cited lack of financial resources, 8% indicated fear of SARS-CoV-2 infection, 2% pointed to being in quarantine, and 1% reported hospitalization.

During the pandemic, 22% of respondents experienced involuntary teeth clenching during the day, and 13% reported teeth grinding, while 70% did not experience either of these symptoms.

In a detailed question about specific symptoms occurring during the pandemic: 43% of respondents reported no complaints, while the most frequently reported issue was loss of dental fillings (21%), followed by tooth sensitivity (20%), tooth chipping (19%), increased facial muscle tension (12%), and temporal headaches (12%).

Additionally, 9% of respondents reported pain in the temporomandibular joint area, and the least common issue—reported by 5—was tooth wear.

Nearly half of the participants were unable to determine whether their dietary habits changed during the pandemic, 28% believed their habits had changed (12% definitely yes, 16% rather yes), while 25% stated they had not changed (8% definitely not, 17% rather not).

In terms of daily nutrition, the majority of respondents (77%) reported cooking their own meals and preparing healthy snacks. Ten percent paid no attention to what they ate, 9% consumed processed ready-to-eat food, and 2% each reported ordering takeaway or consuming fast food.

An increase in snack consumption was observed by 34% of respondents (9% definitely yes, 25% rather yes), while 26% did not observe an increase (17% rather not, 9% definitely not), and 40% were unsure.

A clear majority of respondents—74—stated they paid attention to meal quality (49% rather yes, 25% definitely yes); 12% did not pay attention to quality, and 14% were unsure.

Increased alcohol consumption during the pandemic was reported by 24% of respondents (21% rather more, 3% definitely more), 24% were unsure, and 52% denied any increase (18% rather not, 34% definitely not).

When asked about the use of other stimulants during the pandemic, 15% reported increased use, 16% were unsure, 35% did not observe increased use, and 34% stated the question was not applicable to them.

To enable a more detailed data analysis, responses to six selected survey questions were compared.

A comparative analysis of “Frequency of check-up visits before the outbreak of the pandemic” and “Frequency of check-up visits after the outbreak of the pandemic” showed that the percentage of respondents who underwent dental check-ups once a year remained nearly unchanged (before: 34% vs. 35% after). In the group of those who had dental check-ups once every six months, the percentage decreased by 6% after the pandemic (before 32% vs. 26% after). The largest increase was observed in the group of individuals visiting the dentist less frequently than once every two years, where the percentage rose by 7 percentage points (before 23% vs. 30% after).

When comparing “Tooth brushing time before the pandemic” and “Tooth brushing time after the outbreak of the pandemic,” no significant differences were observed.

The comparison of answers concerning brushing duration revealed differences in the range of 1–3%. The largest increase was noted in the group brushing for 2–3 min, while a decrease was observed in the group brushing for less than 1–2 min (Figure 3).

Before the outbreak of the pandemic, more than half (54%) of the respondents rated their oral hygiene as good and 2% as bad. Similar proportions were found in the groups rating their oral hygiene as very good (21%) or moderate (23%).

The vast majority (69%) of respondents declared that their oral hygiene habits did not change during the pandemic. Approximately one-fifth (23%) observed an improvement in oral hygiene, while the smallest group (8%) reported a decline in their oral hygiene practices.

Across all age groups, most respondents reported no change in their oral health. However, the study showed that oral health among individuals under the age of 30 was significantly more likely (*p* < 0.05) to remain unchanged compared to older participants, while oral health in the 31–50 age group was significantly more likely to improve (Table 2). This pattern indicates that dental health was more stable (less likely to deteriorate or improve) among younger adults under 30 years relative to the middle age group (Table 2).

Due to the extremely small sizes of the primary and lower secondary education categories, they were merged with the secondary education category. Regardless of education level, oral hygiene habits, according to respondents’ answers, most often remained unchanged during the pandemic. The observed differences were minor and therefore considered statistically insignificant (*p* > 0.05). The analysis thus showed that oral hygiene during the pandemic was not significantly associated with the respondents’ level of education (Table 3).

For the purposes of analysis, the categories “definitely yes” and “definitely no” regarding changes in dietary habits were combined with the categories “rather yes” and “rather no,” respectively.

Both women and men most frequently reported being unable to determine whether their dietary habits had changed during the pandemic. The observed differences were statistically insignificant (*p* > 0.05). Therefore, it was concluded that changes in dietary habits during the pandemic were not significantly associated with the sex of the respondents (Figure 4).

Statistically insignificant differences (*p* > 0.05) were also found in the frequency of dental visits, the number of daily tooth brushings, and tooth brushing time before and during the pandemic (Table 4).

The COVID-19 pandemic had a noticeable negative impact on the hygiene and health habits of the surveyed population. The analysis showed a decline in the number of regular dental check-ups during the pandemic. Many individuals experienced dental problems during the pandemic, including teeth clenching. Dietary habits changed, with increased snack and alcohol consumption and more frequent use of stimulants. Following the pandemic, awareness of the importance of oral health increased, as reflected in a greater willingness to return to regular dental visits.

The proposed hypothesis regarding the negative impact of the pandemic and lockdown on dietary, health-related and hygiene habits was confirmed by the study findings.

## 4. Discussion

Over the three years of the pandemic, people’s lives around the world changed significantly, with each of the billions of individuals affected to varying degrees. Daily life, work, and education required ongoing adaptation to a rapidly shifting reality [32]. Government-imposed restrictions aimed to limit and slow the spread of COVID-19, resulting in changes that impacted everyday functioning, including eating habits, physical activity, and even sleep [33]. Many working individuals were subjected to lockdowns and forced into remote work, which, over time, began to affect not only their mental and physical health but also oral health, directly influencing daily hygiene and dietary routines. The findings presented from the population of the West Pomeranian Voivodeship in Poland provide an additional contribution to the limited body of research on health-related behaviors during the pandemic in other European countries. The data obtained may offer valuable insights for planning post-pandemic oral health strategies, as well as for designing preventive and educational frameworks in the event of future pandemics. A detailed analysis of specific survey questions on changes in dietary habits—such as the frequency of snack consumption and the use of stimulants—allowed the collected data to be compared with findings from other studies exploring similar topics.

### 4.1. Consumption of Sweet and Salty Snacks

In the present study, 34% of respondents reported an increase in daily snack consumption—A result comparable to that of Ferrante et al., who noted a 45% increase. Similar trends have been observed by other researchers, who reported a rise in snack consumption, particularly of sweet snacks [34,35,36,37,38]. Lorkova et al. found an increased intake of sweets, dark chocolate, and other snacks, with the rise more pronounced among women than men [39]. Likewise, Sidor et al. reported that 43.5% of participants ate more during lockdown, while 51.8% snacked more frequently between meals [40]. Czeczenek-Lewandowska et al. also observed a statistically significant increase in the consumption of sweets, snacks, and grain products both before and during the pandemic, although the patterns varied across different population groups [41].

Both the present findings and those of other studies showing elevated snack consumption or grazing identified chronic stress caused by isolation and the need to pass time as key contributing factors [41,42,43,44]. England et al. also confirmed an increase in snack and sweet intake during lockdown (from 8.6% to 14.9%), attributing it directly to reduced social interaction due to remote work and isolation [45]. A further rise in sweet snack consumption was reported by Hatipoğlu Palaz et al., who noted increased intake among both children and their parents. Over time, this trend may have serious implications, including a heightened risk of lifestyle-related diseases such as type II diabetes and obesity [46,47].

### 4.2. Changes in Eating Habits

In the present study, 74% of respondents reported changes in their eating habits, including increased attention to meal quality. Similarly, Paltrinieri et al. found that 33.5% of participants became more mindful of healthy eating [48]. In a study of the Slovak population, Lorkova et al. observed a deterioration in eating habits in 22.88% of men and 28.26% of women. Moreover, 30.07% of men and 38.13% of women showed a tendency to overeat. The researchers also recorded a significant increase in the consumption of fresh fruits, vegetables, nuts, and homemade bread [39].

A notable finding was the sharp rise in honey consumption during the pandemic, attributed to its perceived health-promoting properties [39]. Respondents also reported more frequent consumption of home-cooked meals and a decline in fast food intake. Among both men and women, an upward trend was observed in the consumption of sweets and snacks [39]. Similar conclusions were drawn by Błaszczyk-Bębenek et al., who noted a marked decline in dining out and ordering take-away food [49]. According to Sidor et al., 62.3% of participants prepared their own meals at home [40]. In a study conducted among the Spanish population, Navarro-Perez et al. observed an increase in the consumption of fresh foods and a decrease in processed food intake, which they linked to more time spent at home and more opportunities to cook [50].

Various authors have suggested that the increase in home meal preparation was partly driven by efforts to limit contact with others—including delivery personnel—as a way of reducing infection risk. Moreover, the benefits of consuming healthy food and maintaining a balanced diet rich in vitamins and trace elements such as selenium and zinc may offer protective effects against severe COVID-19 and related complications [51].

### 4.3. Alcohol Consumption

In the present study, 24% of respondents reported more frequent alcohol consumption—a trend echoed in other studies [34,36,45,48,49,52,53]. In research by Paltrinieri et al., 12.5% of participants reported increased alcohol intake, while 12.6% reported a decrease. In the study by Paszyńska et al., only 17.3% of respondents stated that they did not consume alcohol [37,48]. Sidor et al. found that 77% of respondents did not observe any increase in their alcohol consumption, while 14.6% reported drinking more. The increase was particularly pronounced among individuals with alcohol dependence, 64% of whom reported higher consumption, compared to 14% of non-dependent individuals [40].

A study by Fernandes et al. conducted in the Portuguese population also found a 16% rise in alcohol consumption during the pandemic. The increase varied by gender, occupational status, and pre-pandemic drinking habits [54]. Rehm et al. reported a significant overall rise in alcohol intake during the pandemic, including a sixfold increase among men [55]. Souza et al. similarly observed increased alcohol consumption, though limited to low-alcohol beverages [53]. According to Rehm et al., in regions where a decline in alcohol intake was observed, it was largely due to government-imposed restrictions that limited access to alcohol, classified as a non-essential good, and led to the closure of establishments where alcohol was typically sold and consumed [55].

A comparison between the present study and other international studies reveals a concerning trend of increased alcohol consumption in many countries during lockdown periods. This phenomenon may be attributed to high psychological burden, difficulties in coping with stress, or underlying socioeconomic factors. It would be valuable to investigate whether these trends were temporary or have become persistent, as well as to explore in greater depth the underlying causes and motivations behind such behaviors.

### 4.4. Stimulant Use and Cigarette Smoking

In the present study, 15% of respondents reported an increased use of stimulants containing nicotine, such as cigarettes or snus. Ferrante et al. specifically identified cigarette smoking among stimulants, noting that 30% of smokers increased their daily cigarette consumption [34]. Other researchers reported more modest changes: Cicero et al. found that 1.7% of smokers smoked more frequently, while 2.2% reported a decrease. In the study by Paltrinieri et al., 7.7% of respondents indicated an increase in smoking, while 4.1% reported a reduction [34,48,52].

In a study conducted among the Turkish population, Keles et al. observed a decline in smoking frequency among 31.6% of both men and women and identified a significant correlation between smoking habits and marital status [56]. According to Sidor et al., 45.2% of respondents reported more frequent smoking, with the largest increase seen among full-time employees [40].

Khamees et al. found that smokers were more likely to engage in unhealthy behaviors, including increased nighttime eating (50.2%), more frequent consumption of fast food, and reduced sleep duration. Respondents attributed increased smoking to stress, fear of job loss, and financial difficulties. The authors linked these patterns to broader factors such as socioeconomic status, lifestyle, and cultural context [57].

Prolonged stress, along with alcohol and nicotine dependence, may increase the risk of SARS-CoV-2 infection [56,58]. During lockdowns, stress affected different groups in varying degrees; however, the predominant response was a rise in addictive behaviors and an increase in frequency of use.

### 4.5. Dental Visits

In the present study, 37% of respondents reported a need for dental care. All of them sought treatment, and 95% received it. Concern about SARS-CoV-2 infection was reported by 38%, while 20% specifically feared contracting the virus during a dental visit. Among those who discontinued regular dental check-ups, 8% cited fear of infection, and 21% pointed to financial constraints (unpublished data).

By comparison, Keles et al. reported that 21.7% of participants required a dental appointment, but only 25.6% actually sought care. Reasons for avoiding dental visits included reluctance to leave home due to the virus (58.3%), fear of infection (44.8%), and viewing healthcare facilities as too risky during the pandemic (40.5%). A considerable proportion—19—did not receive dental treatment despite seeking it, while 16% mentioned financial barriers [56].

In the present study, 20% of respondents expressed concern about the risk of infection in dental offices. Similarly, Sari et al. reported that 50.4% of respondents hesitated to visit the dentist compared to the pre-pandemic period, and 75.6% believed that dental clinics posed a high risk of SARS-CoV-2 transmission [38]. In the study by Paszyńska et al., 52.9% of participants had a dental appointment in 2020 (during the COVID-19 pandemic), while 25% were afraid to schedule a visit [37].

A comparison of findings indicates that the primary differences lie in how individuals responded to the need for dental care. Results from the Polish population suggest greater awareness of the risks associated with delaying treatment and a stronger tendency to actively seek dental services. Particularly notable is the high percentage of Turkish respondents who, despite seeking care, did not receive dental treatment [56].

It is important to highlight the long-term consequences of neglecting dental treatment, which often result in significantly higher financial and time-related burdens compared to preventive care. Limited access to dental services or the discontinuation of regular dental visits may lead to substantial deterioration in oral health, including the progression of dental caries and periodontal diseases, as well as a decrease in the early detection of oral and craniofacial cancers. These findings underscore the need for education and increased public awareness of the critical role of oral health prevention—particularly the importance of regular check-ups, the use of teleconsultations, and educational campaigns promoting oral health.

### 4.6. Oral Hygiene

In the present study, 71% of respondents stated that the pandemic had not affected their oral health, while 69% reported no change in their oral hygiene practices. Similarly, Caramida et al. in the Romanian population observed no significant shift in toothbrushing frequency, though brushing time among healthcare professionals increased by 5.2%. Faria et al. reported that 24% of participants increased the frequency of brushing during the pandemic [59,60]. In contrast, Pinzan-Vercelino et al. found that participants brushed their teeth less frequently and paid less attention to oral hygiene, with a 6.8% increase in the number of individuals unconcerned about the appearance of their smile [61]. Sari et al. reported that 41.1% of respondents began brushing more regularly, while 32.49% altered their brushing habits—9.9% increased their use of oral hygiene products, whereas 7.3% used them less regularly [37,59,60,61].

Guerreiro et al. noted a significant decline in the proportion of individuals brushing 2–3 times daily, from 81.2% before the pandemic to 64.2% during it. In the present study, brushing duration increased during the pandemic, but the change was not statistically significant [62].

Hatipoğlu Palaz et al., who surveyed both adults and their children, found no major differences in brushing time before and after the pandemic. However, they observed a significant change in parental brushing frequency: the proportion of parents brushing fewer than seven times per week rose from 35.5% to 43.2%. The findings suggest that during the pandemic, parents tended to focus more on their children’s oral health than their own [46].

Research conducted in Turkey confirmed the limited impact of the pandemic on oral hygiene habits, consistent with the present study. According to Keles et al., 71.6% of respondents reported no changes in oral hygiene routines, closely aligning with the 69% in this study. Both studies also observed a modest increase in attention to oral care—Keles et al. found an increase in brushing frequency among 20.6% of participants, while in the present study, 23% reported improved oral hygiene during the pandemic [56].

Guerreiro et al. also supported the present findings, noting that a majority of participants relied on manual toothbrushes for daily oral care (68% in their study vs. 88.6% in the present one) [62]. Hatipoğlu Palaz et al. similarly observed frequent use of manual toothbrushes among both children and their parents [46].

In the present study, 50% of respondents reported regular use of dental floss. Guerreiro et al. recorded an 11.7% decrease in flossing frequency during the pandemic compared to pre-pandemic levels, which participants partly attributed to limited access to stores during lockdowns [62]. Hatipoğlu Palaz et al. found no significant change in flossing behavior before and after the pandemic [46]. Keles et al. observed a 7.1% decrease in floss use during the pandemic, while 9.4% of respondents reported an increase; 83.5% noted no change. Overall, 28.3% of respondents used dental floss regularly, including 63.6% of healthcare workers. Their study also revealed a strong correlation between floss use and education level: 48.9% of participants with postgraduate education used dental floss, compared to 17.7% of those with only primary education [56].

In the present study, 49% of participants reported using mouthwash regularly. In the study by Keles et al., 32.8% reported mouthwash use, with a 7.5% decline during the pandemic [56].

It is also important to emphasize systemic and cultural differences in oral hygiene practices. Access to public dental care plays a major role and varies significantly across countries. In well-developed countries, such as the Scandinavian nations, public dental services are widely accessible, whereas in others, such as Poland, high-quality dental treatment is often available only in the private sector. Another crucial factor is health education. In countries like Germany and Japan, where preventive education promotes regular tooth brushing, flossing, and frequent dental check-ups, the proportion of individuals who maintain good oral hygiene and possess health awareness is higher. In contrast, in countries with limited public health campaigns, proper oral hygiene may be underestimated or neglected. Furthermore, in highly developed societies with a strong emphasis on healthy lifestyles, such as the United States, attention to dental aesthetics is often seen as a marker of social status and physical attractiveness. As a result, investing in personal health and appearance is frequently regarded as a priority.

### 4.7. Disorders of the Stomatognathic System

Findings from the present study closely aligned with those reported by Keles et al. concerning dental complaints that emerged during the pandemic. The prevalence of specific symptoms was comparable: tooth fracture and chipping (19% vs. 17.8%), loss of fillings (21% vs. 27.3%), and tooth sensitivity (20% vs. 28.8%) [56].

Regarding bruxism, 12% of participants reported increased facial tension, and another 12% experienced pain in the temporal area—symptoms that may indicate bruxism, even if the individuals were unaware of the condition. Winocur-Arias et al. observed a rise in the number of patients diagnosed with bruxism during the pandemic, with a particularly notable increase among women [63]. Similar trends were reported in studies conducted in Poland and Israel, where initial signs of bruxism often appeared only after the pandemic began. Among individuals already suffering from the condition, symptoms intensified. The main contributing factors cited included elevated stress levels related to illness, isolation, and financial insecurity [64,65].

Further evidence of the growing problem came from Kardes et al., who analysed search engine data to track changes in public concern. While no substantial increase in bruxism-related searches was observed during the initial phase of the pandemic (March–May 2020), a significant rise occurred between May and October 2020. This shift coincided with growing psychological strain resulting from prolonged lockdowns and forced isolation, suggesting that stress-induced bruxism was one of the longer-term consequences of the pandemic [66].

The trends observed by researchers in online searches related to stomatognathic system disorders indicate a clear demand among patients for professional consultations (including teleconsultations) and access to reliable information, which may prove valuable in the context of further patient diagnosis.

Variations across studies may stem from differences in national lockdown policies and the specific phases of the pandemic during which the research was conducted. Nonetheless, a common pattern emerged: increased snack consumption, possibly as a coping mechanism to combat boredom or relieve stress. This is supported by evidence of elevated intake of glucose-rich foods, which, by affecting the hypothalamic-pituitary-adrenal axis, reduce stress response activity. The release of hormones during the consumption of sweet foods is known to promote relaxation and comfort [67,68,69]. However, these behaviours contribute to the development of unhealthy dietary habits, rising BMI, and the continued progression of the obesity epidemic [70].

The relative stability in oral hygiene routines may reflect a high level of health awareness among respondents, as well as a deliberate effort to maintain oral health in response to reduced access to dental services, high infection rates, and worsening economic conditions during the early phases of the pandemic [71].

### 4.8. Fear of SARS-CoV-2 Infection

In the present study, 38% of respondents reported fear of contracting SARS-CoV-2, including 13.5% of whom were students. Similarly, a study by Špiljak et al. conducted among Croatian students found that 25% experienced anxiety related to receiving a positive SARS-CoV-2 test result [72]. High levels of concern were also reported by Xu et al., who observed that among various population groups in China, healthcare professionals expressed the greatest fear of infection (71.8%), while students reported a high level of concern as well (39.2%) [73]. In a study conducted by Gambhir et al. among the Indian population, as many as 69% of respondents expressed fear regarding potential complications resulting from COVID-19 infection [74].

These findings illustrate notable differences in the reported levels of fear concerning SARS-CoV-2 infection, which may be influenced by both cultural context and professional background. The highest levels of anxiety were observed among healthcare workers, who, due to their occupational exposure, faced a higher risk of infection and likely possessed greater awareness of the clinical course of the disease and its potential complications.

Over time, individuals and societies began to adjust to the realities of pandemic life. This adaptability highlights the remarkable human capacity to function in times of prolonged crisis. The literature lacks detailed studies linking the impact of the pandemic with the incidence and progression of dental caries and periodontal diseases, particularly in relation to limited access to dental care during the pandemic period.

### 4.9. The Strength and Limitations

The study followed a standardized procedure and was preceded by a pilot phase, which allowed us to refine and clarify the questionnaire items. A key strength is the diversity of the sample, which included patients from both academic and private clinical settings, as well as the comprehensiveness of the questionnaire itself.

Nonetheless, several limitations should be acknowledged. The sample size (n = 100) limits the generalizability of the findings, and the non-random recruitment strategy may have introduced selection bias. Self-reported changes in health status during the pandemic are also subject to recall and response biases. Moreover, participants’ perceptions may have varied depending on which phase of the pandemic they experienced.

Emotional factors such as fear, uncertainty, and stress may have further influenced responses. Future research should build on these findings by using larger, more representative samples and employing a longitudinal design to assess how health-related behaviors evolve over time.

## 5. Conclusions

### 5.1. Conclusions

During the pandemic, many countries lacked recommendations for patients regarding oral hygiene practices, dietary guidelines, and instructions on when to seek dental care, as well as preventive measures aimed at maintaining oral well-being and preserving healthy eating habits for as long as possible. The COVID-19 pandemic had a clearly negative impact on health behaviors, dietary habits, and oral hygiene practices in the studied population. A decrease in the frequency of routine dental visits was observed during the pandemic, despite an increased awareness of the importance of oral health after its conclusion. The pandemic significantly altered dietary habits, including an increase in the consumption of snacks, alcohol, and the use of stimulants. Although oral hygiene routines largely remained unchanged, some respondents reported either deterioration or improvement in these practices, indicating varied behavioral responses. Psychological stress and lockdown-related restrictions likely contributed to the occurrence of parafunctions such as teeth clenching and bruxism. These findings highlight the need to implement targeted public health actions, including dental education, nutritional counseling, and preventive strategies, as well as information on access to care, in order to mitigate the long-term negative effects of lifestyle changes associated with the pandemic.

### 5.2. Recommendations

Given the limited duration of the study period encompassing the pandemic, there is a clear need for further longitudinal research to evaluate shifts in treatment requirements and health-related behaviors in subsequent years. Such investigations would help to discern which habits formed during the pandemic have been maintained and which have diminished following its resolution. Additionally, it is essential to assess the extent of complications and the progression of dental caries and periodontal diseases attributable to the neglect of routine dental care. Findings from these studies would provide valuable insights for the planning and delivery of dental services, the development of preventive strategies, and the design of targeted educational interventions, ultimately improving the alignment of care provision with patient needs. Potential approaches to mitigate these issues may include the implementation of telemedicine services, remote dental consultations, and systematic reminders for regular dental check-ups and preventive treatments such as professional dental cleanings. Moreover, it is warranted to initiate preventive measures aimed at enhancing awareness of the adverse health consequences associated with neglecting routine dental examinations. It is worth considering the development of an easily accessible tool for patients that would respond to their current needs in a given crisis situation, serving as both support and a source of useful information.

## Figures and Tables

**Figure 1 epidemiologia-06-00067-f001:**
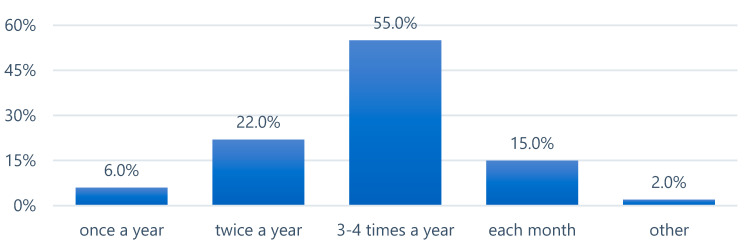
Frequency of toothbrush or toothbrush head replacement.

**Figure 2 epidemiologia-06-00067-f002:**
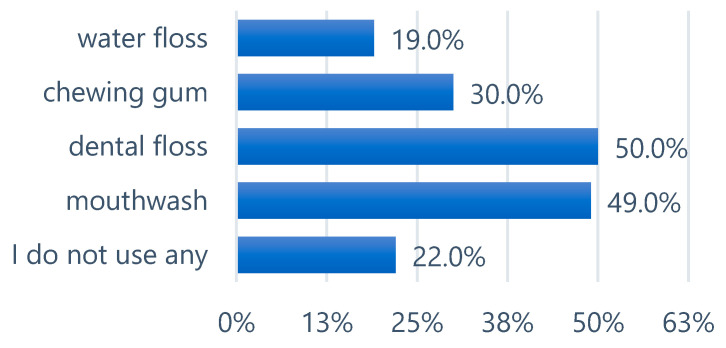
Other additional oral hygiene products used by respondents.

**Figure 3 epidemiologia-06-00067-f003:**
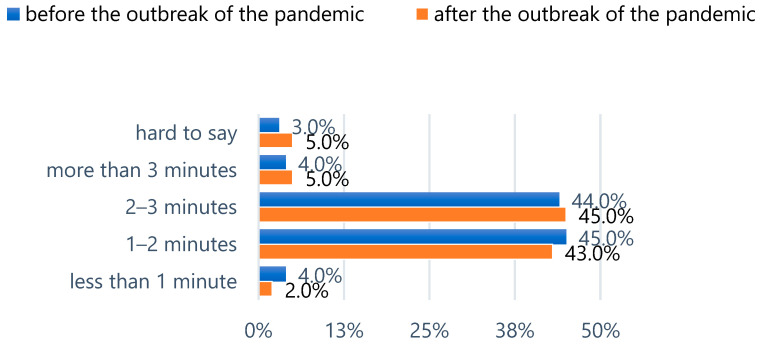
Tooth brushing time before and during the pandemic.

**Figure 4 epidemiologia-06-00067-f004:**
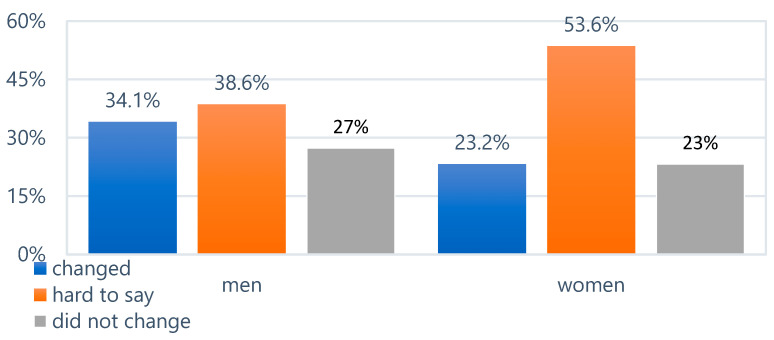
Changes in dietary habits during the pandemic.

**Table 1 epidemiologia-06-00067-t001:** Characteristics of the study group.

Age	N	%
<30 years	38	38.0%
31–50 years	33	33.0%
>50 years	29	29.0%
**Sex**		**%**
female	56	56.0%
male	44	44.0%
**Health Status**	**N**	**%**
very good	48	48.0%
good	42	42.0%
moderate	10	10.0%
**Place of Residence**	**N**	**%**
above 300,000 inhabitants	70	70.0%
100,000–300,000 inhabitants	9	9.0%
20,000–100,000 inhabitants	18	18.0%
below 20,000 inhabitants	1	1.0%
rural areas	2	2.0%
**Education**	**N**	**%**
primary	3	3.0%
lower secondary	3	3.0%
secondary	54	54.0%
college	40	40.0%
**Employment Status**	**N**	**%**
employed	65	65.0%
maternity leave	1	1.0%
retirement	7	7.0%
disability pensioner	2	2.0%
student	22	22.0%
unemployed	3	3.0%

**Table 2 epidemiologia-06-00067-t002:** Self-assessed oral health status by age group. (A) Self-assessed oral health status by age group. Post-hoc test.

			Age Groups			Test Result
			<30 Years	31–50 Years	>50 Years	
Dental health during the pandemic	rather deteriorated	N	1	6	5	*p* = 0.033, V = 0.22
		%	2.6%	18.2%	17.2%	
	did not change	N	31	16	18	
		%	81.6%	48.5%	62.1%	
	improved	N	6	11	6	
		%	15.8%	33.3%	20.7%	
Total		N	38	33	29	
		%	100.0%	100.0%	100.0%	
(**A**)						
**Comparison**			***p* (Fisher)**	***p* (Adjusted)**	**Cramér’s *V***	**Post-hoc Power**
<30 years vs. 31–50 years			0.010	**0.029**	0.366	0.80
<30 years vs. >50 years			0.101	0.152	0.273	0.50
31–50 years vs. >50 years			0.544	0.544	0.152	0.17

Note. N group size; *p*—statistical significance; *V*—Cramér’s measure of association. Pairwise Fisher’s exact tests with adjusted *p*-values (likely via FDR correction). Cramér’s V indicates effect size (0.10 = small, 0.30 = moderate, 0.50 = large). Analyses are post-hoc to the overall Fisher’s exact test (*p* = 0.033, N = 100).

**Table 3 epidemiologia-06-00067-t003:** Oral hygiene habits by education level.

			Education		Test Result
			Secondary or Lower	College	
Oral hygiene practices during the pandemic	deteriorated	N	4	4	*p* = 0.794, *V* = 0.08
		%	6.7%	10.0%	
	did not change	N	41	28	
		%	68.3%	70.0%	
	improved	N	15	8	
		%	25.0%	20.0%	
Total		N	60	40	
		%	100.0%	100.0%	

N—group size; *p*—statistical significance; *V*—Cramér’s measure of association.

**Table 4 epidemiologia-06-00067-t004:** Comparison of health-related behaviors before and during the pandemic.

			Before the Pandemic	During the Pandemic	Test Results
Frequency of check-up visits	every six months	N	32	26	χ^2^ = 1.76 df = 3 *p* = 0.624, *V* = 0.09
		%	32.0%	26.0%	
	once a year	N	34	35	
		%	34.0%	35.0%	
	once every two years	N	11	9	
		%	11.0%	9.0%	
	less frequently	N	23	30	
		%	23.0%	30.0%	
Tooth brushing frequency	once a day or less	N	20	21	χ^2^ = 0.16 df = 2 *p* = 0.921 *V* = 0.03
		%	20.0%	21.0%	
	twice a day	N	63	59	
		%	63.0%	59.0%	
	three times a day or more	N	17	18	
		%	17.0%	18.0%	
Tooth brushing duration	up to 2 min	N	49	45	χ^2^ = 0.19 df = 1 *p* = 0.663 φ = 0.03
		%	50.5%	47.4%	
	more than 2 min	N	48	50	
		%	49.5%	52.6%	

χ^2^—test statistic; df—degrees of freedom; N—group size; *p*—statistical significance; φ—phi measure of association; *V*—Cramér’s measure of association.

## Data Availability

The datasets generated during and analyzed during the current study are not publicly available due to restrictions in the participants’ signed consent.

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
