# Peer review of "The Impact of the COVID-19 Pandemic Caused by SARS-CoV-2 on Hygiene, Health, and Dietary Habits: A Survey-Based Study"

_epidemiologia, 2025, doi:10.3390/epidemiologia6040067_

Round 1
Reviewer 1 Report
Comments and Suggestions for Authors
- please add the validity and reliability of the questionnaire.
- please mention the possible disability and medical history of participants.
- add a paragraph about the limitations and bias in discussion.
- please add a link or QRcode for free access to the questionnaire.
- add a section for recommendation further studies to compare the treatment needs in following years after pandemy.
Author Response
Please see the attachment.
Thank you
Best regards

Reviewer 2 Report
Comments and Suggestions for Authors
The impact of the COVID-19 pandemic caused by SARS-CoV-2 on hygiene, health, and dietary habits: a survey-based study
Reviewer Report
First and foremost, the manuscript needs to be prepared in accordance with the journal’s formatting guidelines.
Abstract
The abstract of the study does not adequately reflect the scientific depth, systematic structure, and linguistic sophistication required for an academic publication. The research objective is vague and presented in general terms, while the methodology section lacks essential details regarding study design, sample selection, questionnaire validity, and data analysis methods. The results are superficial, with no inclusion of statistical data or numerical values. The conclusion fails to highlight the study’s contribution or its clinical and societal relevance, and appears repetitive. It is recommended that the abstract be rewritten in a more structured manner, supported by objective data, and aligned with an appropriate scientific tone.
Introduction
Introduction section provides a comprehensive overview of the global impacts of the SARS-CoV-2 pandemic on healthcare systems, lifestyle, dietary habits, and especially oral health. The inclusion of the “Aim of the study” heading and its statement clarifies that the study seeks to assess the impact of the COVID-19 pandemic on health behaviors, oral hygiene practices, and dietary habits. However, the study’s rationale and clinical contribution remain insufficiently emphasized, leaving the research’s originality and the gap in the literature unclear. Moreover, the hypothesis is not stated, and the introduction does not fully explain why the study is necessary or how it adds to existing knowledge. While the aim is outlined, it is presented in a general manner without detailing specific research questions or potential clinical implications. Therefore, it is recommended that the introduction be further developed to explicitly highlight the study’s significance, its unique contribution, and to clearly state the hypothesis to enhance the overall clarity and scientific value of the manuscript.
Materials and Methods
Materials and methods section clearly describes the cross-sectional survey design, the sample group, inclusion criteria, ethical approval, and informed consent processes in detail. The content, structure, and administration of the questionnaire are thoroughly outlined, with clear information on the scope of questions (socio-demographic data, hygiene, dietary habits, and pre- and during-pandemic comparisons). Additionally, methodological rigor is demonstrated by addressing data privacy and respondent anonymity. However, there is no explanation regarding how the sample size was calculated or selected, which is a limitation concerning the study’s statistical power and generalizability. While measurement parameters are clearly defined, there is no information about the reliability of the measurements, such as repeatability or error analysis. Details about the researchers’ experience or standardization of the procedure are also missing. The statistical analyses used (Chi-square, Fisher’s exact test, ANOVA, Mann–Whitney, Kruskal–Wallis) and their application criteria are explained comprehensively, and software versions are specified, ensuring transparency. Overall, the methodology is satisfactory in covering fundamental aspects; however, including details on sample size calculation, measurement reliability, and researcher expertise would enhance the scientific validity of the manuscript.
Discussion
This section addresses changes in individuals’ lifestyles, dietary habits, alcohol and tobacco use, oral hygiene, and dental health behaviors during the pandemic with a detailed and extensive literature review. The study’s findings are compared with various international studies in the literature, and similarities and differences are discussed comprehensively. However, the discussion does not clearly highlight the study’s unique contribution or rationale, nor does it sufficiently emphasize the clinical and public health significance of the findings. There is no dedicated section on the study’s strengths and limitations, and methodological constraints or potential weaknesses related to the sample are not discussed. Concrete suggestions for future research or guidance on existing gaps are also absent. Additionally, original comments on the practical applications of the findings are limited, and a more critical approach in interpreting the results is needed. Overall, while the section provides a thorough literature review, it is recommended that the discussion be structured to be more focused, critical, and include clear recommendations to strengthen the scientific contribution of the study.
Conclusion
Conclusion section briefly summarizes the main results of the study but does not provide the reader with concrete conclusions based on the study’s results. Additionally, the section largely constitutes a repetition of the results and lacks an in-depth evaluation of the clinical or societal significance of the findings or how these findings should be interpreted in the context of the study’s objectives. It is recommended that the conclusion be revised to include stronger, data-driven inferences and to directly address the study’s aims in a more analytical and interpretative manner.
Author Response
Please see the attachment.
Thank you!
Best regards

Reviewer 3 Report
Comments and Suggestions for Authors
Dear authors, this manuscript is well-organized and thoroughly examines the effects of the COVID-19 pandemic on dietary, hygiene, and oral health practices within a Polish cohort. Below are my suggestions to enhance its quality:
1) Although the manuscript offers extensive data, the discussion would greatly benefit from a more in-depth interpretation of the clinical implications of these results—especially regarding dental public health. For example, the reduction in routine dental check-ups during the pandemic may lead to long-term repercussions such as a higher incidence of undiagnosed caries, progression of periodontal disease, and decreased patient adherence to preventive dental care. Furthermore, stress-related behaviors like bruxism, as noted in the study, can result in irreversible harm if not promptly addressed. Exploring how dental professionals and public health entities could alleviate these risks in the post-pandemic period—through targeted education, recall systems, or teledentistry—would enhance the practical significance of the manuscript.
2) The research offers important insights into shifts in oral hygiene practices and dental attendance. Nevertheless, in the discussion section, these results are somewhat eclipsed by the emphasis on dietary behaviors. Considering the journal's focus and the critical role of oral health in the broader context of COVID-19-related behaviors, the authors should aim for a more equitable analysis. For instance, more emphasis could be placed on trends regarding toothbrush replacement, the use of supplementary hygiene products (such as floss and mouthwash), and anxiety related to dental appointments. These findings could be more thoroughly compared to international literature to underscore cultural and systemic variations in oral health practices.
3) The manuscript makes a general reference to "stimulants" without detailing the specific substances that fall under this category. To enhance clarity and uphold scientific rigor, it is advisable for the authors to delineate which products are included—such as caffeine, nicotine, or energy drinks—and to clarify how respondents understood the question.
4) The study should place its findings within a wider European biomedical context and provide an additional viewpoint on oral health awareness and behavioral changes among biomedical students during the COVID-19 pandemic, given that approximately one-quarter of the participants in this study were students. It is crucial to assess their self-reported intentions to return to regular dental care and to provide comparative insights into health-related knowledge and decision-making within a similar European cultural framework (please refer to: https://pubmed.ncbi.nlm.nih.gov/39851604/)
Kind regards!
Author Response

(The authors gave the same response as above.)

Round 2
Reviewer 2 Report
Comments and Suggestions for Authors
There are still some misunderstandings regarding the study:
First of all, please revise your manuscript according to the journal’s template. You can download the relevant template from the following link: https://www.mdpi.com/journal/epidemiologia/instructions (Please download the Microsoft Word template file).
Sample size calculation determines the minimum number of participants required to be included in the study and the statistical power (as a percentage) of the study with that minimum sample size. This calculation should be performed using G*Power or a similar program. The calculation is based on data from a previous similar study. Please revise your manuscript accordingly.
Lastly, in scientific writing, it should be clear that the revisions requested by the reviewers have been made. Therefore, all changes should be made using track changes mode or by highlighting the revised text in a different color. In its current form, it is not possible to determine whether the manuscript has been revised or not.
Author Response
Please see the attachment.
Thank you.
Kind regards
Aleksandra Wdowiak-Szymanik

Reviewer 3 Report
Comments and Suggestions for Authors
Dear authors, you have significantly enhanced the quality of your manuscript. After final proofreading and putting it the MDPI Epidemiologia's template, in my opinion, it should be accepted for publication. Best regards!
Author Response
Please see the attachment.
Thank you!
Kind regards
Aleksandra Wdowiak-Szymanik
